# A Novel Strategy of US3 Codon De-Optimization for Construction of an Attenuated Pseudorabies Virus against High Virulent Chinese Pseudorabies Virus Variant

**DOI:** 10.3390/vaccines11081288

**Published:** 2023-07-27

**Authors:** Mengwei Xu, Yiwei Wang, Yamei Liu, Saisai Chen, Laixu Zhu, Ling Tong, Yating Zheng, Nikolaus Osterrieder, Chuanjian Zhang, Jichun Wang

**Affiliations:** 1National Research Center of Engineering and Technology for Veterinary Biologicals, Institute of Veterinary Immunology and Engineering, Jiangsu Academy of Agricultural Sciences, Nanjing 210014, China2019207062@stu.njau.edu.cn (S.C.); 19970021@jaas.ac.cn (J.W.); 2GuoTai (Taizhou) Center of Technology Innovation for Veterinary Biologicals, Taizhou 225300, China; 3Jiangsu Co-Innovation Center for Prevention and Control of Important Animal Infectious Diseases and Zoonoses, Yangzhou 225009, China; 4Jiangsu Key Laboratory for Food Quality and Safety-State Key Laboratory Cultivation Base of the Ministry of Science and Technology, Jiangsu Academy of Agricultural Sciences, Nanjing 210014, China; 5College of Veterinary Medicine, Nanjing Agricultural University, Nanjing 210095, China; 6Institut für Virologie, Freie Universität Berlin, 14163 Berlin, Germany

**Keywords:** pseudorabies virus, virulence, safety, immunogenicity, codon de-optimization

## Abstract

In this study, we applied bacterial artificial chromosome (BAC) technology with PRV^ΔTK/gE/gI^ as the base material to replace the first, central, and terminal segments of the US3 gene with codon-deoptimized fragments via two-step Red-mediated recombination in *E. coli* GS1783 cells. The three constructed BACs were co-transfected with gI and part of gE fragments carrying homologous sequences (gI+gE’), respectively, in swine testicular cells. These three recombinant viruses with US3 codon de-optimization ((PRV^ΔTK&gE-US3deop−1^, PRV^ΔTK&gE-US3deop−2^, and PRV^ΔTK&gE-US3deop−3^) were obtained and purified. These three recombinant viruses exhibited similar growth kinetics to the parental AH02LA strain, stably retained the deletion of TK and gE gene fragments, and stably inherited the recoded US3. Mice were inoculated intraperitoneally with the three recombinant viruses or control virus PRV^ΔTK&gEAH02^ at a 10^7.0^ TCID_50_ dose. Mice immunized with PRV^ΔTK&gE-US3deop−1^ did not develop clinical signs and had a decreased virus load and attenuated pathological changes in the lungs and brain compared to the control group. Moreover, immunized mice were challenged with 100 LD_50_ of the AH02LA strain, and PRV^ΔTK&gE-US3deop−1^ provided similar protection to that of the control virus PRV^ΔTK&gEAH02^. Finally, PRV^ΔTK&gE-US3deop−1^ was injected intramuscularly into 1-day-old PRV-negative piglets at a dose of 10^6.0^ TCID_50_. Immunized piglets showed only slight temperature reactions and mild clinical signs. However, high levels of seroneutralizing antibody were produced at 14 and 21 days post-immunization. In addition, the immunization of PRV^ΔTK&gE-US3deop−1^ at a dose of 10^5.0^ TCID_50_ provided complete clinical protection and prevented virus shedding in piglets challenged by 10^6.5^ TCID_50_ of the PRV AH02LA variant at 1 week post immunization. Together, these findings suggest that PRV^ΔTK&gE-US3deop−1^ displays great potential as a vaccine candidate.

## 1. Introduction

Pseudorabies is an acute infectious disease caused by a highly virulent and contagious herpesvirus, pseudorabies virus (PRV). Although the natural host is the pig, it can also infect many different mammals, including cattle, goats, cats, dogs, sheep, and wild animals, in which it causes often-fatal disease of the central nervous system [1,2]. Despite immunization with classic vaccines, such as Bartha-K61, there has been ongoing virus evolution with apparent antigenic variation in PRV in China since 2011, which has resulted in several outbreaks of swine pseudorabies in pig farms [3,4,5]. Thus, vaccine development is necessary to combat newly arising variants.

In our previous study, we constructed a PRV variant TK/gE double gene deletion strain (PRV^ΔTK&gE-AH02^), which was safe for 1-day-old PRV antibody-positive piglets and 4- to 5-week-old PRV-negative piglets. Moreover, the strain provided robust protection to a recent PRV variant. However, it showed some pathogenicity in 1-day-old PRV antibody-negative piglets [6]. Consequently, further reducing the virulence of this particular PRV strain while preserving the immunogenicity seemed necessary.

The basic amino acids (except methionine and tryptophan) are often encoded by two or more synonymous codons. Further, evolution has led to different codon usage frequencies in different species or various tissues of the same species [7]. In this context, codon de-optimization has emerged as a tool for replacing synonymous codons in the existing virus sequences with suboptimal codons [8]. The rationale of the technique is that suboptimal codons are used less frequently in the host cell, reducing the mRNA stability, affecting translation efficiency, and thus significantly reducing protein expression levels [8,9,10]. Moreover, it should be noted that codon de-optimization does not result in changes to the amino acid sequence of the viral proteins, which retains the same antigenic epitopes as the wild-type virus and thus remains completely immunogenic [9,11]. Owing to these aspects, codon de-optimization-based modification of viruses to attenuate virulence has become a popular trend to produce vaccine candidates. Several attenuated viruses have been generated using various codon de-optimization strategies [12,13,14,15,16]. We here focused on the US3 gene, which encodes a serine/threonine kinase. Previous studies indicated that the US3 gene was dispensable for virus growth in cells but was a critical virulence factor in vivo [17]. Motivated by these aspects, we constructed three recombinant viruses with US3 codon de-optimization based on a PRV TK/gE double gene deletion strain (PRV^ΔTK&gE-AH02^) by replacing partial synonymous codons with suboptimal codons in the first, central, and terminal segments of the US3 gene with reference to the porcine codon usage frequency table and codon adaptation index (CAI). We then systematically evaluate their safety and immune potency in mice and piglets.

## 2. Materials and Methods

### 2.1. Materials

#### 2.1.1. Virus and Cells

The PRV variant AH02LA (CGMCC No. 10891) and its PRV AH02LA TK/gE double gene deleted mutant (PRV^ΔTK&gE-AH02^) strain were identified and preserved in the laboratory. Swine testicular (ST) cells (CVCC: CL27) were purchased from the American Type Culture Collection (ATCC, Manassas, VA, USA) and cultured in Dulbecco’s Modified Eagle Medium (DMEM, Thermo Fischer Scientific, Waltham, MA, USA) containing 2% or 10% fetal bovine serum (FBS, Gibco, Waltham, MA, USA) with 100 IU/mL penicillin (Sigma-Aldrich, St. Loius, MO, USA) and 100 μg/mL streptomycin (Sigma-Aldrich) in an environment of 5% CO_2_ at 37 °C.

#### 2.1.2. Plasmids and Strains

T-Vector pMD19 (Simple) plasmid was purchased from TaKaRa Co. Ltd. (Dalian, China). *E. coli* GS1783 cells (a kind gift of Dr. Gregory Smith, Northwestern University) containing BAC^PRVΔTK/gE/gI&K+^ or BAC^PRVΔTK/gE/gI^ were prepared and stored in our laboratory. The pmKate2-N vector was purchased from Evrogen (Moscow, Russia).

#### 2.1.3. Main Reagents

Trypsin was purchased from Thermo Fischer Scientific (Waltham, MA, USA). DNA and RNA extraction kits were purchased from Omega (Bienne, Switzerland). T4 DNA ligase, LA Taq polymerase, DH5α receptor cells, the plasmid extraction kit, proteinase K, and restriction endonucleases (*Dpn* I and *Kpn* I) were purchased from Dalian Bao Biological Engineering Co; Ltd. (Dalian, China). The Universal Column Genome Extraction Kit was purchased from Beijing Kangwei Century Co. Ltd. (Beijing, China). The 2×TSINGKE^®^ Master quantitative polymerase chain reaction (qPCR) Mix (SYBR Green I) was purchased from Tsingke Biotechnology Co. Ltd. (Nanjing, China). Lipofectamine^®^ 3000 was purchased from Invitrogen (Waltham, MA, USA).

#### 2.1.4. Test Animals

The 4- to 6-week-old healthy Institute of Cancer Research (ICR) female mice (18–22 g) were purchased from Nanjing Qinglong Mountain Animal Breeding Farm (Nanjing, China) and reared in the mice room of the North Animal House of Jiangsu Academy of Agricultural Sciences (Nanjing, China). The 1-day-old PRV-negative piglets and sows were self-raised in the laboratory and reared in the same pens of a pig farm of Jiangsu Academy of Agricultural Sciences. The piglets, aged 28–35 days old, were self-raised and fed twice daily (at 8:00 and 17:00) in the pig farm of Jiangsu Academy of Agricultural Sciences. It should be noted that all the test pigs were negative for PRV, swine fever virus, cerebrospinal virus, porcine reproductive and respiratory syndrome virus, porcine parvovirus, and porcine circovirus type 2.

### 2.2. Methods

#### 2.2.1. Construction of Bacterial Artificial Chromosomes (BACs) with US3 Codon De-Optimization

The primer sequences used in this study are shown in Table 1. The primers were synthesized by the Tsingke Biotechnology Co. Ltd. (Nanjing, China). The codon de-optimized US3 (US3^deop^-1, US3^deop^-2, US3^deop^-3 in the Appendix A) was obtained after the codon de-optimization of the US3 gene into the first, central, and terminal segments with reference to the porcine codon usage frequency table and CAI. The recombinant plasmids, T-US3^deop^-1, T-US3^deop^-2, and T-US3^deop^-3, were constructed by ligating the synthesized US3^deop^-1, US3^deop^-2, and US3^deop^-3 with the linearized PMD19-T simple vectors. Then, plasmids T-US3^deop^-1, T-US3^deop^-2, and T-US3^deop^-3 were digested with restriction endonucleases *Cla* I, *Btg* I, and *BspE* I, respectively. The target fragments were ligated with kanamycin resistance gene, respectively, and then transformed into DH5α cells to obtain recombinant plasmids. Finally, PCR and sequencing were performed to identify whether the kanamycin resistance gene was correctly ligated in recombinant plasmids.

H1-US3^deop^-1-Kan-H2 (US3^deop^-1-Kan with homologous arms), H1-US3^deop^-2-Kan-H2 (US3^deop^-2-Kan with homologous arms), and H1-US3^deop^-3-Kan-H2 (US3^deop^-3-Kan with homologous arms) DNA were initially amplified with primers of H1-US3^deop^-1-Kan-H2 F/R, H1-US3^deop^-2-Kan-H2 F/R, H1-US3^deop^-3-Kan-H2 F/R (Table 1). After digestion with *Dpn* I, three PCR products were, respectively, electroporated into *E. coli* GS1783 with BAC^PRVΔTK/gE/gI^. Approximately 100–150 monoclonal colonies were grown on each plate. There was no significant difference among 3 constructs. Further, single colonies were picked to extract the BAC DNAs. Then, the extracted BAC DNAs were used as a template for PCR identification and sequencing with primers, i.e., US3^deop^-1-Kan ide F/R, US3^deop^-2-Kan ide F/R, and US3^deop^-3-Kan ide F/R (Table 1). These three identified BAC DNAs and the control BAC^PRVΔTK/gE/gI^ were digested by *Kpn* I. Then, the restriction fragment length polymorphism (RFLP) analysis was performed and compared with the predicted profiles. Finally, the positive clones were named BAC^PRVΔTK/gE/gI-US3deop−1&K+^, BAC^PRVΔTK/gE/gI-US3deop−2&K+^, and BAC^PRVΔTK/gE/gI-US3deop−3&K+^, respectively.

The obtained BAC^PRVΔTK/gE/gI-US3deop−1&K+^, BAC^PRVΔTK/gE/gI-US3deop−2&K+^, and BAC^PRVΔTK/gE/gI-US3deop−3&K+^ were subjected to a second step of Red recombination to remove the kanamycin resistance genes. The BAC DNA was initially extracted using the double plate resistance screening approach and further characterized with PCR, and subsequently sequenced using primers PRV US3 check F/R, respectively. The three PCR-positive BAC DNA were selected and digested by *Kpn* I for RFLP analysis. Finally, the cloned strains that matched with the predicted profiles were named BAC^PRVΔTK/gE/gI-US3deop−1^, BAC^PRVΔTK/gE/gI-US3deop−2^, and BAC^PRVΔTK/gE/gI-US3deop−3^, respectively.

#### 2.2.2. Acquisition of Recombinant Viruses with US3 Codon De-Optimization

The DNA of PRV^ΔTK&gE-AH02^ was used as a template to amplify the gI and partial gE gene fragments with homologous arms (gI+gE’), using the primers PRV gI+gE’ F/R. Then, the gI+gE’ fragments were co-transfected with BAC^PRVΔTK/gE/gI-US3deop−1^, BAC^PRVΔTK/gE/gI-US3deop−2^, and BAC^PRVΔTK/gE/gI-US3deop−3^ DNA, respectively, on ST cells using Lipofectamine^®^ 3000 following the manual of supplier. The mini-F sequences of BAC^PRVΔTK/gE/gI-US3deop−1^, BAC^PRVΔTK/gE/gI-US3deop−2^, and BAC^PRVΔTK/gE/gI-US3deop−3^ were replaced with gI+gE’ fragments. After 24 h of transfection, the culture medium was discarded and supplemented with DMEM containing 10% FBS and 0.5% methylcellulose for 24–48 h. Then, the viruses with no green fluorescence emission were picked under the excitation of Ultraviolet (UV) light at a wavelength of 488 nm and inoculated with fresh ST cells. After picking for several rounds, the DNAs from the three purified viruses were extracted, and the gE gene was identified through PCR and sequencing with a pair of primers (PRV gE site check F/R). Finally, the adequately identified strains were denoted PRV^ΔTK&gE-US3deop−1^, PRV^ΔTK&gE-US3deop−2^, and PRV^ΔTK&gE-US3deop−3^, respectively. A schematic diagram of the construction of PRV^ΔTK&gE-US3deop−1^ as an example is shown in Figure 1.

#### 2.2.3. Detection of US3 Gene Expression from the Virus Background

To detect the US3 gene expression level from the virus background, PRV^ΔTK&gE-US3deop−1^, PRV^ΔTK&gE-US3deop−2^, PRV^ΔTK&gE-US3deop−3^, and PRV^ΔTK&gE-AH02^, as well as parental AH02LA strains, were inoculated with a multiplicity of infection (MOI) of 10 into a monolayer of ST cells. ST cells were harvested at different times (2, 6, and 10 h) after infection. Then, total RNA was extracted and subjected to reverse transcription. After reverse transcription, the expression of the US3 gene in different virulent strains was detected using fluorescence qPCR. The primers are shown in Table 1.

#### 2.2.4. Detection of US3 Protein Expression of Eukaryotic Plasmids with Codon De-Optimized US3

US3 primers, i.e., US3^deop^-1 inred F, US3^deop^-2/3 inred F, and US3^deop^-1/2/3 in red R with *EcoR* I sites, were initially designed according to the pmKate2-N vector sequence (Table 1). Further, the PRV^ΔTK&gE-AH02^, PRV^ΔTK&gE-US3deop−1^, PRV^ΔTK&gE-US3deop−2^, and PRV^ΔTK&gE-US3deop−3^ were used as templates, respectively, to amplify US3 fragments with homologous vector sequences (h1-US3-h2, h1-US3^deop^-1-h2, h1-US3^deop^-2-h2, and h1-US3^deop^-3-h2). Then, the pmKate2-N vector was digested by *EcoR* I.

The h1-US3-h2, h1-US3^deop^-1-h2, h1-US3^deop^-2-h2, and h1-US3^deop^-3-h2 were ligated with the pmKate2-N vector, respectively. The ligation system was set as follows: 5 μL of 2×Seamless Cloning Mix, 2 μL of US3 fragment, 1 μL of *EcoR* I enzymatic linearization pmKate2-N vector, and 2 μL of dd-H_2_O, reacted at 50 °C for 30–45 min. Then, 5–10 μL of the cooled recombinant product was transformed into DH5α cells to obtain recombinant plasmids, i.e., pmKate2-N-US3, pmKate2-N-US3^deop^-1, pmKate2-N-US3^deop^-2, and pmKate2-N-US3^deop^-3. The recombinant plasmids were identified using PCR and sequencing with primers pmKate2-US3 inred F/R.

The plasmid transfection was performed using Lipofectamine^®^ 3000 following the manual of the supplier. Initially, ST cells were seeded in the 6-well plates, and the transfection process was performed after the growth rate attained a confluence of 70–90%. Then, 10 µg of pmKate2-N-US3, pmKate2-N-US3^deop^-1, pmKate2-N-US3^deop^-2, and pmKate2-N-US3^deop^-3 was added to the plates for transfection, respectively, and they continued to be cultured in the incubator maintained at 37 °C and 5% CO_2_. Finally, the expression level of the red fluorescent protein of each plasmid was observed under an inverted fluorescence microscope at 24 h post-transfection.

#### 2.2.5. Cytopathic Effect

To determine the characteristics of cytopathic effects (CPE), PRV^ΔTK&gE-US3deop−1^, PRV^ΔTK&gE-US3deop−2^, PRV^ΔTK&gE-US3deop−3^, PRV^ΔTK&gE-AH02^, and parental AH02LA strains were inoculated into the 6-well plates seeded with freshly grown monolayers of ST cells at a MOI of 0.01. Cells were incubated with a cell maintenance solution to 0.6 mL at 37 °C. After 1 h of incubation, the inoculum was removed and DMEM containing 2% FBS was added. Further, CPE were observed under an inverted microscope at 24, 48, and 72 h post-infection, respectively.

#### 2.2.6. Plaque Assay

PRV^ΔTK&gE-US3deop−1^, PRV^ΔTK&gE-US3deop−2^, PRV^ΔTK&gE-US3deop−3^, and PRV^ΔTK&gE-AH02^, along with parental AH02LA, were inoculated into the 6-well plates seeded with fresh ST cells at a MOI of 0.01. DMEM was added as a cell maintenance solution to 0.6 mL and incubated at 37 °C for 1h. Further, the inoculum was removed and a cell maintenance medium containing 1% low melting point agarose and 2% FBS was added. After solidifying the medium, the plate was further incubated for 24 h. A total of 100 images of each virus were taken under the same doubling microscope for each plaque, whose surface area was determined by Image J software (LOCI, University of Wisconsin, Madison, WI, USA). The plaque areas of other strains were compared with that of the parental AH02LA strain, which was set to 100%. All experiments were repeated thrice, independently.

#### 2.2.7. Virus Growth Kinetics

Initially, PRV^ΔTK&gE-US3deop−1^, PRV^ΔTK&gE-US3deop−2^, PRV^ΔTK&gE-US3deop−3^, and PRV^ΔTK&gE-AH02^, as well as parental AH02LA strains, were inoculated on ST cells at a MOI of 0.01, respectively. At 6, 12, 24, 36, 48, 60, and 72 h post-infection, the ST cells were collected with the supernatant mixture. After freeze–thawing thrice at −80 °C and 37 °C, the cell debris was removed using centrifugation, and the viral fluid was harvested. The viral TCID_50_ were determined on ST cells according to the Reed–Muench method. All experiments were independently replicated thrice.

#### 2.2.8. Genetic Stability Test

The genetic stability test of the modified strains was performed as stated below. Initially, PRV^ΔTK&gE-US3deop−1^, PRV^ΔTK&gE-US3deop−2^, and PRV^ΔTK&gE-US3deop−3^ were transmitted continuously on ST cells for up to 20 generations (F20). gE, TK, and the recoded US3 genes in PRV^ΔTK&gE-US3deop−1^, PRV^ΔTK&gE-US3deop−2^, and PRV^ΔTK&gE-US3deop−3^ were identified via PCR and sequencing with primers PRV ΔTK check F/R, PRV gE site check F/R, and PRV US3 check F/R, respectively.

#### 2.2.9. Pathogenicity and Immunological Experiments in Mice

Healthy ICR female mice (*n* = 128) aged 4 to 6 weeks were randomly divided into 12 groups of 8 mice each. Mice were subcutaneously inoculated with 0.2 mL of PRV^ΔTK&gE-AH02^, PRV^ΔTK/gE-US3deop−1^, PRV^ΔTK/gE-US3deop−2^, or PRV^ΔTK/gE-US3deop−3^ (10^7.0^ TCID_50_/0.2 mL, 10^6.0^ TCID_50_/0.2 mL, and 10^5.0^ TCID_50_/0.2 mL), respectively. The negative control group was inoculated with 0.2 mL of DMEM. Finally, the clinical signs and mortality rate of mice were monitored daily for 14 days.

The healthy ICR female mice (*n* = 20) aged 4–6 weeks were randomly divided into 4 groups of 5 mice each. Further, the mice were intraperitoneally injected with 0.2 mL of PRV^ΔTK/gE-US3deop−1^, PRV^ΔTK/gE-US3deop−2^, PRV^ΔTK/gE-US3deop−3^, or PRV^ΔTK&gE-AH02^ (10^7.0^ TCID_50_/0.2 mL), respectively. At 5 days post-infection, all mice were sacrificed and major organs, i.e., brain and lung samples, were collected. The tissue samples were prepared by weighing 0.1 mg of samples separately and placing them in the grinding beads. After grinding, the samples were centrifuged. Then, the supernatant was used to extract DNA, and the PRV gB gene was used as the target. The viral load was quantified with real-time quantitative PCR using primers gB F/R. Notably, the gB standard curve was used for the linear regression analysis of copy numbers. In addition, histopathological sections of the brain and lung tissues were prepared, and the changes were observed.

Based on our previous study [18], after 2 weeks of immunization in the 10^7.0^ TCID_50_ dose group, mice were administered with 100 LD_50_ of PRV AH02LA strain. Further, the clinical signs and mortality rate of mice were monitored daily for 14 days after the challenge. At the end of the experiment, all surviving mice were sacrificed and disposed of harmlessly.

#### 2.2.10. Pathogenicity and Immunological Experiments in Piglets

Briefly, 1-day-old healthy piglets (*n* = 15), negative for PRV, were randomly divided into 3 groups of 5 piglets each. The piglets in the immunized group were intramuscularly inoculated with 1 mL of PRV^ΔTK&gE-AH02^ (10^6.0^ TCID_50_/mL), PRV^ΔTK/gE-US3deop−1^ (10^6.0^ TCID_50_/mL), and DMEM, respectively. High-dose PRV vaccines were used to evaluate their safety in 1-day-old piglets in clinical applications [6,18]. Further, body temperatures, clinical signs, and the mortality rate of piglets in different treatment groups were recorded daily for 14 days. Finally, serum samples were collected from piglets at 7, 14, and 21 days post-inoculation and tested for neutralizing antibody index. Then, 100 μL of each serum sample (heat inactivated for 30 min at 56 °C) was mixed with an equal volume of virus (AH02LA) at a different dilution. The neutralization indexes were calculated as the TCID_50_ of piglet serum in the test group divided by the TCID_50_ of negative serum.

Piglets aged 28–35 days old (*n* = 15) were randomly divided into 3 groups of 5 piglets each and reared in isolated environments in group cages before treatment. The piglets were intramuscularly inoculated with 1 mL of PRV^ΔTK&gE-AH02^ (10^5.0^ TCID_50_/mL), PRV^ΔTK/gE-US3deop−1^ (10^5.0^ TCID_50_/mL), and DMEM, respectively. At 1 week post-immunization, all piglets were administered with 2 mL of PRV AH02LA strain (10^6.5^ TCID_50_/2 mL) via nasal drip based on our previous study [6]. After challenging, the body temperatures, clinical signs, and mortality rate of piglets were recorded daily for 14 days. Nasal swab samples were collected daily from 0 to 14 days post challenge to detect virus shedding. After shaking and freeze–thaw cycles (−80 °C and 37 °C), samples were centrifuged (10,000 rpm) and the supernatants were used to determine the viral titers.

### 2.3. Statistical Analysis

The data were represented as mean ± standard deviation (SD) and analyzed with one-way analysis of variance (ANOVA) followed by Tukey’s test using GraphPad Prism 5 software (GraphPad Software, Inc., La Jolla, CA, USA), considering a *p*-value less than 0.05 as statistically significant. * represents *p* < 0.05, ** indicates *p* < 0.01, *** signifies *p* < 0.001, and *p* > 0.05 refers to a not-significant result.

## 3. Results

### 3.1. Generation of Recombinant BAC with US3 Codon De-Optimization

Initially, the codon de-optimized US3 containing the kanamycin resistance gene was subjected to the two-step Red-mediated homologous recombination in E. coli GS1783 containing BAC^PRVΔTK/gE/gI^, and the kanamycin resistance gene was knocked out. Further, the three BACs were identified using the double-plate resistance screening, and PCR and sequencing. Then, BAC DNA was cleaved with *Kpn* I and subjected to RFLP analysis. It was observed that the predicted PRV (GenBank: KM061380.1) profiles were essentially identical (Figure 2A). Compared to BAC^PRVΔTK/gE/gI-US3deop−1&K+^, BAC^PRVΔTK/gE/gI-US3deop−2&K+^, BAC^PRVΔTK/gE/gI-US3deop−3&K+^, BAC^PRVΔTK/gE/gI-US3deop−1^, BAC^PRVΔTK/gE/gI-US3deop−2^, and BAC^PRVΔTK/gE/gI-US3deop−3^ possessed an additional band of 5843 bp size and a missing band of 6881 bp size.

### 3.2. Construction and Identification of Recombinant Viruses with US3 Codon De-Optimization

Typically, BAC^PRVΔTK/gE/gI-US3deop−1^, BAC^PRVΔTK/gE/gI-US3deop−2^, and BAC^PRVΔTK/gE/gI-US3deop−3^ DNA were co-transfected with gI+gE’ fragments in the ST cells, respectively. Successful co-transfection could be observed under UV excitation at a wavelength of 488 nm (Figure 2B). After three rounds of picking and plating, purified recombinant viruses with US3 codon de-optimization (PRV^ΔTK&gE-US3deop−1^, PRV^ΔTK&gE-US3deop−2^, and PRV^ΔTK&gE-US3deop−3^) were obtained. PCR and its sequencing analyses validated the successful recovery of gI and part of the gE genes). As depicted in Figure 2C, the mRNA expression levels of US3 genes of PRV^ΔTK&gE-US3deop−1^, PRV^ΔTK&gE-US3deop−2^, and PRV^ΔTK&gE-US3deop−3^ were significantly lower than that of PRV^ΔTK&gE-AH02^ at 10 h post-infection (*p* < 0.001). As shown in Figure 2D, recombinant eukaryotic plasmids pmKate2-N-US3, pmKate2-N-US3^deop^-1, pmKate2-N-US3^deop^-2, and pmKate2-N-US3^deop^-3 expressed red fluorescent protein at 24 h post-transfection, and the red fluorescent protein expression levels of pmKate2-N-US3^deop^-1, pmKate2-N-US3^deop^-2, and pmKate2-N-US3^deop^-3 were lower than that of pmKate2-N-US3.

### 3.3. Growth Characteristics of Recombinant Viruses with US3 Codon De-Optimization

Among various strains, the parent AH02LA strain resulted in the most significant number of spots and the largest plaque area at 24 h post-infection. Moreover, cells were fused and shed in large numbers, while the lesions of PRV^ΔTK&gE-AH02^-infected ST cells appeared slightly delayed when compared with those of the parent AH02LA strain. Nevertheless, the plaque morphology showed no significant difference from that of the parent AH02LA strain after 48 h and 72 h of infection. In contrast, PRV^ΔTK&gE-US3deop−1^, PRV^ΔTK&gE-US3deop−2^, and PRV^ΔTK&gE-US3deop−3^ showed smaller plaques, presenting cell fusion and only rarely cell flotation (Figure 3A,B). As observed from the multi-step viral growth curves (Figure 3C), PRV^ΔTK&gE-US3deop−1^, PRV^ΔTK&gE-US3deop−2^, and PRV^ΔTK&gE-US3deop−3^ displayed similar growth kinetics when compared with PRV^ΔTK&gE-AH02^. The peak titers of PRV^ΔTK&gE-US3deop−1^, PRV^ΔTK&gE-US3deop−2^, PRV^ΔTK&gE-US3deop−3^, and PRV^ΔTK&gE-AH02^, as well as the parent AH02LA strains, were 10^7.67^, 10^7.97^, 10^7.75^, 10^7.75^, and 10^8.18^ TCID_50_/mL, respectively.

### 3.4. Safety and Immunogenicity in Mice

Clinical symptoms and the number of deaths were displayed in Table 2. Mice inoculated subcutaneously with 0.2 mL of PRV^ΔTK&gE-US3deop−1^, PRV^ΔTK&gE-US3deop−2^, PRV^ΔTK&gE-US3deop−3^ or PRV^ΔTK&gEAH02^ (10^7.0^ TCID_50_/0.2mL, 10^6.0^ TCID_50_/0.2mL, and 10^5.0^ TCID_50_/0.2mL) showed no significant clinical signs, and all mice survived after inoculation.

In addition, mice were intraperitoneally inoculated with 10^7.0^ TCID_50_ of PRV^ΔTK&gE-US3deop−1^, PRV^ΔTK&gE-US3deop−2^, PRV^ΔTK&gE-US3deop−3^, or PRV^ΔTK&gEAH02^. All mice were sacrificed at 5 days post-inoculation, and the viral copy numbers in the brains and lungs of mice were determined using qPCR. Viral DNA copy numbers in the brain and lungs of PRV^ΔTK&gE-US3deop−1^-inoculated mice were lower than those of PRV^ΔTK&gEAH02^-inoculated mice (*p* < 0.05). PRV^ΔTK&gE-US3deop−2^-inoculated mice showed lower copy numbers in the lungs than PRV^ΔTK&gEAH02^-inoculated mice (*p* < 0.05). However, no significant difference in viral load between the other groups was observed (Figure 4A).

The histopathological analysis of major organs (brain and lung) in mice was performed at 5 days post-inoculation with 10^7.0^ TCID_50_ of PRV^ΔTK/gE-US3deop−1^, PRV^ΔTK/gE-US3deop−2^, PRV^ΔTK/gE-US3deop−3^, or PRV^ΔTK&gE-AH02^. In the PRV^ΔTK/gE-AH02^-inoculated mice, inflammatory cell infiltration in the lungs and brains were observed. In contrast, no significant pathological changes were observed in the brains and lungs of mice inoculated with PRV^ΔTK/gE-US3deop−1^ and PRV^ΔTK/gE-US3deop−2^ (Figure 4B).

In the next experiment, mice were immunized with 10^7.0^ TCID_50_ of PRV^ΔTK&gE-US3deop−1^, PRV^ΔTK&gE-US3deop−2^, PRV^ΔTK&gE-US3deop−3^, and PRV^ΔTK&gEAH02^. At 14 days post-immunization, mice were administered with 100 LD_50_ of PRV AH02LA strain. It was observed that all mice in the control group succumbed to infection at 4 days post-challenge. The survival rates were eventually recorded as 75% for both PRV^ΔTK&gE-US3deop−1^ and PRV^ΔTK&gEAH02^ groups and 50% and 37.5% for PRV^ΔTK&gE-US3deop−2^ and PRV^ΔTK&gE-US3deop−3^ groups, respectively (Figure 4C and Appendix A).

### 3.5. Safety and Immunogenicity in Piglets

Considering the safety and immunogenicity results in mice, we further explored the safety and immunogenicity of PRV^ΔTK&gE-US3deop−1^ in piglets. One-day-old piglets were inoculated intramuscularly with 10^6.0^ TCID_50_ of PRV^ΔTK&gE-US3deop−1^ or PRV^ΔTK&gE-AH02^ and the temperature changes as well as clinical signs were monitored for 14 days (Figure 5A,B and Appendix A). After inoculation with PRV^ΔTK&gE-US3deop−1^, it was observed that two piglets presented a normal body temperature and showed no other substantial clinical symptoms. However, three piglets exhibited a body temperature of 40.0–40.5 °C for 2–3 days, and two piglets showed mild respiratory symptoms. In contrast, after PRV^ΔTK&gE-AH02^ vaccination, four piglets displayed a body temperature of 40–41 °C for 3–6 days, and two piglets developed clinical symptoms, such as sneezing, coughing, and loss of appetite, while the other piglets appeared healthy.

Moreover, serum was collected from piglets at 7, 14, and 21 days post-immunization to detect the neutralizing antibody index (Figure 5C). At 7 days post-immunization, the neutralizing antibody index was low in the PRV^ΔTK&gE-US3deop−1^ and PRV^ΔTK&gE-AH02^ treatment groups, and it subsequently increased progressively. At 14 days post-immunization, the neutralizing antibody index was 50,118 in the PRV^ΔTK&gE-US3deop−1^ treatment group and 72,443 in the PRV^ΔTK&gE-AH02^ treatment group. At 21 days post-immunization, the neutralizing antibody index was 1,698,244 in the PRV^ΔTK&gE-US3deop−1^ group and 2,884,031 in the PRV^ΔTK&gE-AH02^ group. Notably, no significant difference in the neutralizing antibody index was observed between the PRV^ΔTK&gE-AH02^ and PRV^ΔTK&gE-US3deop−1^ groups at 7, 14, and 21 days post-immunization. In future studies, we will also detect the duration of antibodies in piglets immunized with PRV^ΔTK&gE-US3deop−1^. For protective efficacy, all piglets in the challenge control group developed a reduced appetite and sneezing at 2 days post-challenge. At 3 days post-challenge, typical symptoms, such as nasal mucous discharge, salivation, abdominal breathing, depression, and loss of appetite, began to appear. The body temperature of all piglets in the challenge control group reached over 41 °C for 3–6 days. All five piglets in the challenge control group died during the test period. After the AH02LA challenge, no clinical signs or temperature responses were observed in piglets immunized with PRV^ΔTK&gE-US3deop−1^ and PRV^ΔTK&gE-AH02^ (Figure 6A,B and Appendix A).

After the challenge, nasal swabs were collected daily to determine the virus titers in the excreted nasal discharge. It was observed that all piglets in the control group started to shed the virus at 1 day post-challenge, and continued to do so until death. However, no virus was detected in piglets immunized with PRV^ΔTK&gE-US3deop−1^ and PRV^ΔTK&gE-AH02^ (Figure 6C and Appendix A).

## 4. Discussion

Since 2011, a virulent variant of PRV has developed into an epidemic in pig operations in China. It has increasingly been recognized that the traditional Bartha-K61 vaccine showed poor immune protection, lacking the ability to prevent virus shedding [19,20,21]. To deal with the hazardous nature of PRV mutant strains, several PRV gene-deleted and attenuated vaccines have been developed using mutant strains that can improve protection against PRV variant strains, and discriminate between immunized pigs and pigs infected with wild-type virus [6,22,23,24]. Nevertheless, the safety of these attenuated strains for PRV antibody-negative neonatal piglets is particularly perturbing [6]. The further attenuation of these strains while maintaining immunogenicity is necessary to develop a safe and effective live PRV vaccine.

Codon de-optimization has emerged as one of the efficient strategies for the engineering of live-attenuated vaccines in many systems [9,12]. Notably, codon de-optimization is often achieved using suboptimal codons to recode virulence genes. This approach offers many advantages, as stated in the following. Regarding safety, codon de-optimization often relies on introducing many synonymous mutations in virulence genes, leading to the highly efficient attenuation of the virus by reducing viral gene expression [8,25]. Considering excellent immunogenicity, codon de-optimization did not change the amino acid sequence in the viral protein, retaining the same antigenic epitopes as the wild-type virus [8]. In terms of simple operation, the codon de-optimized viruses could be generated rapidly via gene synthesis and reverse genetics, and it is theoretically possible to control the degree of attenuation.

Although a lack of viral protein kinase activity results in reduced virulence, several kinases, including herpesvirus kinase, are often dispensable for growth in cell culture [17,18,26]. The US3 gene encodes a 334–390 amino acid protein serine/threonine kinase, and is a positive regulator of viral replication and pathogenicity [27,28]. Notably, the US3 gene is involved in viral particle formation, cytoskeletal rearrangement, and the escape of multiple host antiviral responses [29,30,31].

In the current study, the US3 gene was divided into three segments, and codon de-optimization was designed for each. It was observed that the protein expression levels of the recombinant eukaryotic plasmid were significantly reduced after the US3 codon de-optimization. Further, the virus growth curve assay showed that the US3 gene de-optimization showed no substantial effect on the virus proliferation. Nevertheless, several differences were evident when compared to the parental virus, such as the delayed time of plaque formation, a rare occurrence of cell fusion and smaller plaques. The genetic stability test revealed that there was no change in the codon de-optimized sequences, heralding a very low possibility of reversion to virulence. The three recombinant viruses with US3 codon de-optimization were tested for pathogenicity and immunogenicity in mice. The results showed that PRV^ΔTK&gE-US3deop−1^ decreased the virus load and attenuated pathological changes in the brain and lung of mice compared with PRV^ΔTK&gE-AH02^. The protection efficiency of PRV^ΔTK&gE-US3deop−1^ was similar to PRV^ΔTK&gE-AH02^ in mice. Further, a piglet safety test showed that PRV^ΔTK&gE-US3deop−1^ was less pathogenic in piglets than PRV^ΔTK&gE-AH02^, producing high levels of neutralizing antibody. Finally, an evaluation of the immunoprotective effect of PRV^ΔTK&gE-US3deop−1^ in piglets showed that it not only provided complete protection, but also prevented virus shedding against the virulent variant AH02LA challenge at 1 week post-immunization. It is clear that levels of antibodies correlate poorly with the decreased virus replication early after infection, and PRV-specific lymphocyte proliferation responses and a rapid influx of T lymphocytes at the site of viral replication play an important role in the clearance of PRV infection [32,33]. Therefore, the PRV-specific cell-mediated immune response may have been related to the prevention of clinical disease and virus shedding early after challenge. Future studies involving cell-mediated immunity analysis are necessary to better understand the mechanisms of immune protection induced by live PRV vaccines.

## 5. Conclusions

In summary, a recombinant virus with US3 codon de-optimization (PRV^ΔTK&gE-US3deop−1^) was successfully constructed. With its high immune efficacy, it could be a suitable vaccine candidate, presenting its potential for eliminating newly arising PRV variants in pig farms. This study also provided a new theoretical basis and technical means for PRV vaccine development.

## Figures and Tables

**Figure 1 vaccines-11-01288-f001:**
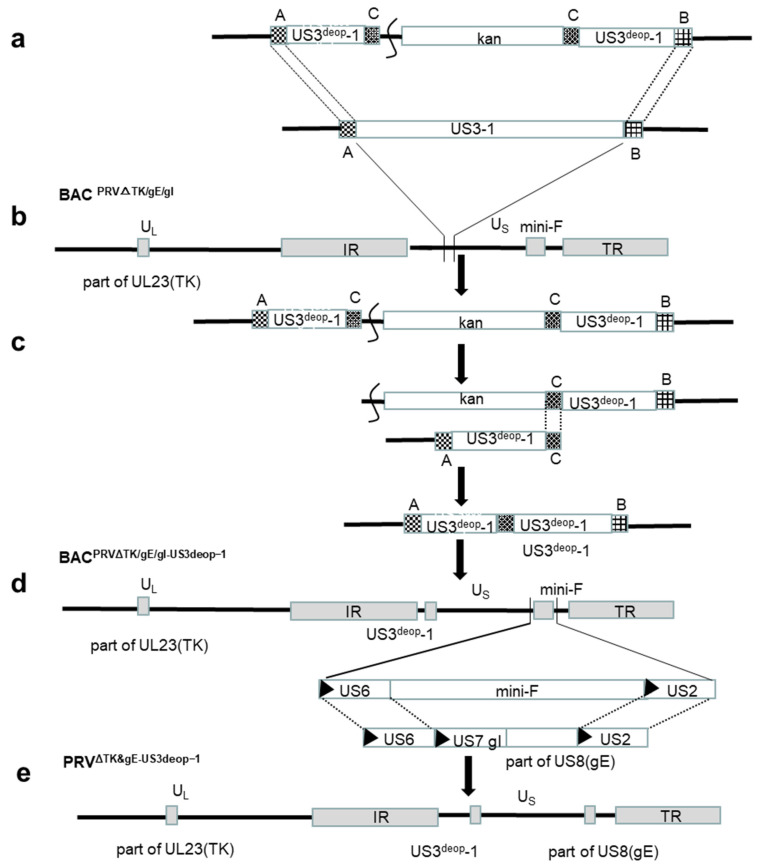
A schematic diagram showing the construction of the recombinant virus (PRV^ΔTK&gE-US3deop−1^) with US3 codon de-optimization. (**a**) A kanamycin resistance gene with a 40 bp homologous sequence (C) was inserted into the *Cla* I restriction site in the T-US3^deop^-1. (**b**) US3^deop^-1-Kan with homologous arms (A and B) amplified with primers of H1-US3^deop^-1-Kan-H2 F/R. US3^deop^-1-Kan was inserted in lieu of US31 of BAC^PRVΔTK/gE/gI^ through the first recombination. (**c**) The kanamycin resistance gene was removed in the second step. (**d**) Homologous recombination was performed to substitute the mini-F sequence with the whole gI gene and part of the gE gene. (**e**) Recombinant virus with US3 codon de-optimization (PRV^ΔTK&gE-US3deop−1^) was obtained.

**Figure 2 vaccines-11-01288-f002:**
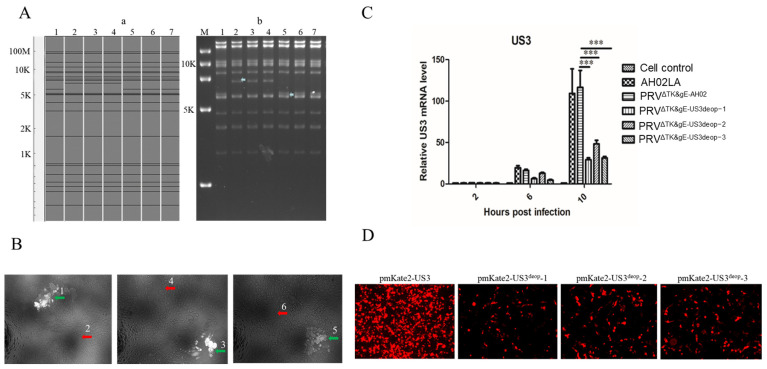
Data presenting the construction and identification of recombinant BACs and recombinant viruses. (**A**) The images present the identification of recombinant BAC with US3 codon de-optimization via RFLP analysis. (**a**) The graph shows the predicted RFLP profile using the PRV ZJ01 strain (GenBank: KM061380.1) as the reference. (**b**) M stands for DNA Marker DL 15,000, 1 stands for BAC^PRVΔTK/gE/gI^, 2 stands for BAC^PRVΔTK/gE/gI-US3deop−1&K+^, 3 stands for BAC^PRVΔTK/gE/gI-US3deop−2&K+^, 4 stands for BAC^PRVΔTK/gE/gI-US3deop−3&K+^, 5 stands for BA ^PRVΔTK/gE/gI-US3deop−1^, 6 stands for BAC^PRVΔTK/gE/gI-US3deop−2^, and 7 stands for BAC^PRVΔTK/gE/gI-US3deop−3^. (**B**) The image shows the plaque of recombinant viruses with US3 codon de-optimization. (1) PRV^ΔTK/gE/gI-US3deop−1^, (2) PRV^ΔTK&gE-US3deop−1^, (3) PRV^ΔTK/gE/gI-US3deop−2^, (4) PRV^ΔTK&gE-US3deop−2^, (5) PRV^ΔTK/gE/gI-US3deop−3^, (6) PRV^ΔTK&gE-US3deop−3^. (**C**) The graph presents different viruses expressing US3 gene mRNA levels. Data are presented as mean ± SD (*n* = 3 in each group). *** signifies *p* < 0.001. (**D**). The image shows the red fluorescent protein expressed by the corresponding recombinant plasmids.

**Figure 3 vaccines-11-01288-f003:**
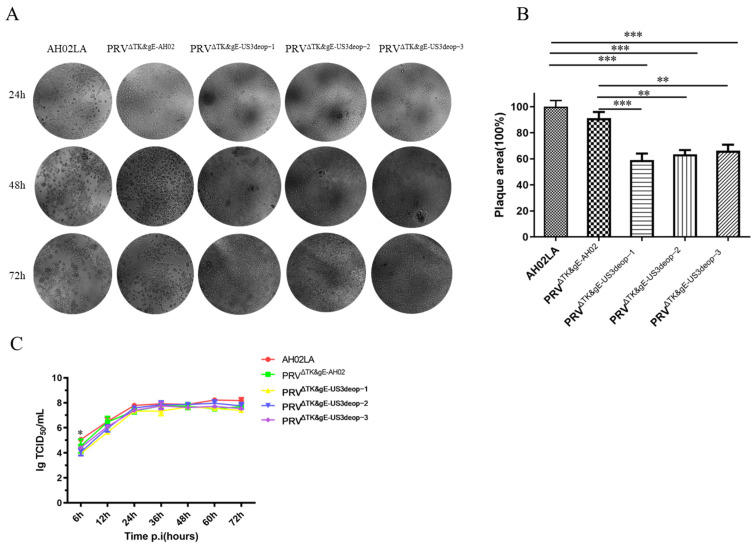
Data presenting the growth characteristics of recombinant viruses with US3 codon de-optimization on ST cells. (**A**) The image shows the morphology of plaques with different virus strains on ST cells at different times. ST cells were infected with PRV^ΔTK&gE-US3deop−1^, PRV^ΔTK&gE-US3deop−2^, PRV^ΔTK&gE-US3deop−3^, and PRV^ΔTK&gE-AH02^, as well as parental AH02LA strains at a MOI of 0.01. At 24, 48, and 72 h post-infection, CPE were observed under an inverted microscope. (**B**) The graph presents plaque size statistics of different virus strains on ST cells at 24 h post-infection. The plaque size induced by the parental AH02LA strain was set TO 100%. Data are presented as mean ± SD (*n* = 3 in each group). ** represents *p* < 0.01, and *** signifies *p* < 0.001. (**C**) The graph indicates the multi-step growth kinetics of different virus strains. ST cells are infected with PRV^ΔTK&gE-US3deop−1^, PRV^ΔTK&gE-US3deop−2^, PRV^ΔTK&gE-US3deop−3^, and PRV^ΔTK&gE-AH02^, as well as parental AH02LA strains at a MOI of 0.01. At 6, 12, 24, 36, 48, 60, and 72 h post-infection, the virus was collected and titrated on ST cells. Data are presented as mean ± SD (*n* = 3 in each group). * indicates *p* < 0.05.

**Figure 4 vaccines-11-01288-f004:**
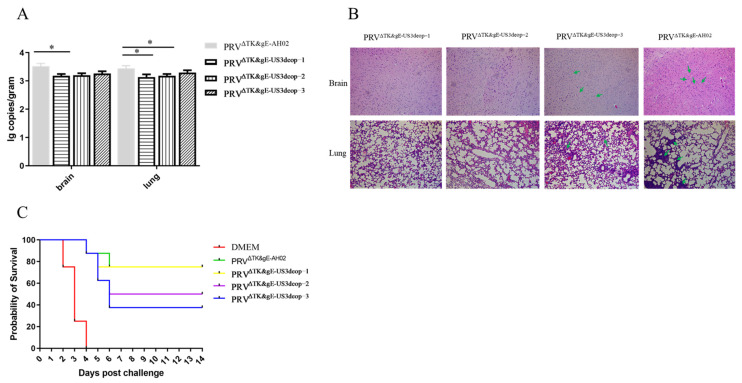
Data validating the evaluation of safety and immunogenicity of recombinant viruses with US3 codon de-optimization in mice. (**A**) The graph presents the quantification of the viral DNA load in different tissue samples of mice at 5 days post-infection. Data are presented as mean ± SD (*n* = 5 in each group). * indicates *p* < 0.05. (**B**) The image shows the histopathology examination of the lungs and brains of mice at 5 days post-infection. Arrows show inflammatory cell infiltration in the brain and lung. (**C**) The graph presents the percentage survival of mice inoculated with different virus strains after challenge (*n* = 8 in each group).

**Figure 5 vaccines-11-01288-f005:**
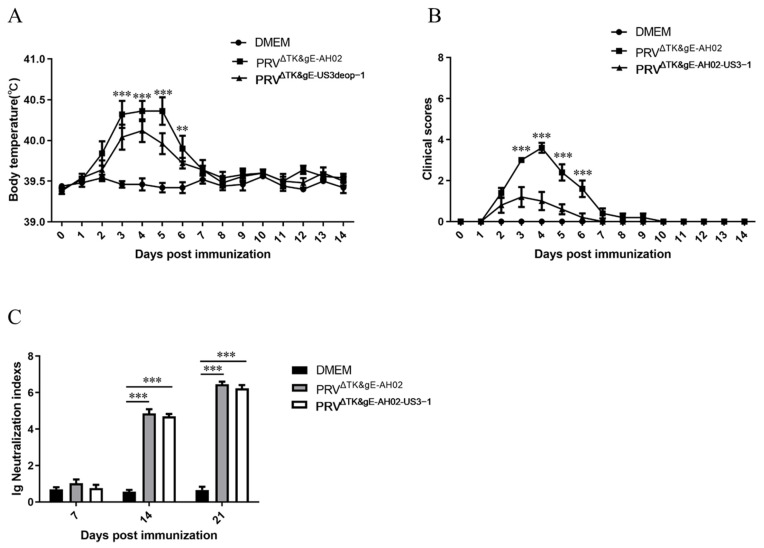
Data validating the evaluation of safety and immunogenicity of recombinant viruses with US3 codon de-optimization in piglets. (**A**) The graph presents the rectal temperatures of piglets after vaccination. Data are presented as mean ± SD (*n* = 5 in each group). ** represents *p* < 0.01, and *** signifies *p* < 0.001. (**B**) The image shows the clinical scores of piglets after vaccination. Data are presented as mean ± SD (*n* = 5 in each group). *** signifies *p* < 0.001. (**C**) The graph presents the serum neutralizing antibody index of piglets at 7, 14, and 21 days post-vaccination. Data are presented as mean ± SD (*n* = 5 in each group). *** signifies *p* < 0.001.

**Figure 6 vaccines-11-01288-f006:**
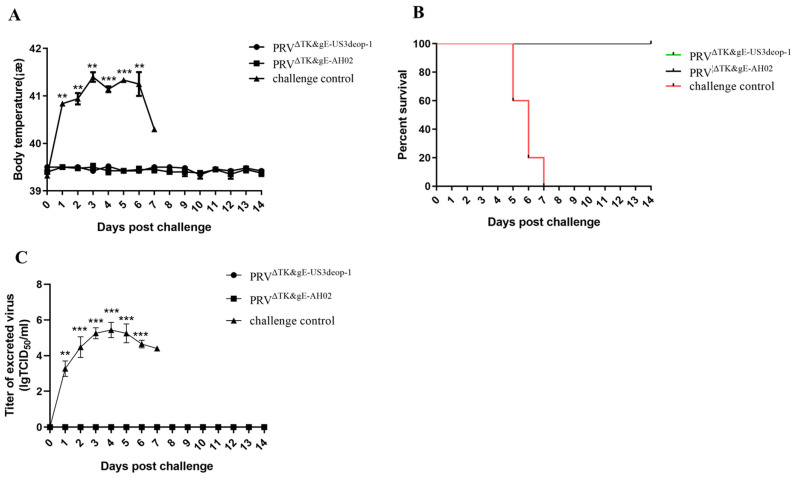
Data indicating the protective efficacy of PRV^ΔTK&gE-US3deop−1^ against the highly virulent strain AH02LA challenge in piglets. (**A**) The image presents the rectal temperatures of piglets after the PRV AH02LA challenge. Data are presented as mean ± SD (*n* = 5 in each group). ** represents *p* < 0.01, and *** signifies *p* < 0.001. (**B**). The graph shows the survival rate of piglets after the PRV AH02LA challenge (*n* = 5 in each group). (**C**). The image indicates the virus excretion of piglets after the PRV AH02LA challenge. Data are presented as mean ± SD (*n* = 5 in each group). ** represents *p* < 0.01, and *** signifies *p* < 0.001.

**Table 1 vaccines-11-01288-t001:** A summary showing the primers used for genetic manipulation, PCR, real-time PCR or sequencing.

Primers	Sequence (5′–3′)	Aim
PRV US3^deop^-1-Kan F	CCCATCGATGGCGATGGCGATAGCAGGATGACGACGATAAGTAGGGATAAC	Amplification of US3^deop^-1-Kan
PRV US3^deop^-1-Kan R	CCCATCGATGATGATTTCATCATCGCTGGGTAATGCCAGTGTTACAACCA	
PRV US3^deop^-2-Kan F	TCCCCCGGGGCCCGGTGACGTGTCTGGGATGACGACGATAAGTAGGGATAAC	Amplification of US3^deop^-2-Kan
PRV US3^deop^-2-Kan R	TCCCCCGGGCGAGCATCTGCTTCAGGGGGTAATGCCAGTGTTACAACCA	
PRV US3^deop^-3-Kan F	CCCATCGATCTCATCCGCGCCCTCAGGATGACGACGATAAGTAGGGATAAC	Amplification of US3^deop^-3-Kan
PRV US3^deop^-3-Kan R	CCCATCGATCAGATGCATCTCCCCGTTGGGTAATGCCAGTGTTACAACCA	
US3^deop^-1-Kan ide F	CCGATGAAATCCTGTATTCG	Identification of US3^deop^-1-Kan
US3^deop^-1-Kan ide R	GCTCATAACACCCCTTGTAT	
US3^deop^-2-Kan ide F	GACGTGTCTGGTCCTGCCGCATTT	Identification of US3^deop^-2-Kan
US3^deop^-2-Kan ide R	AATCGCGGCCTCGAGCAAGA	
US3^deop^-3-Kan ide F	GGGGTGCATCCCGAAGAATTTC	Identification of US3^deop^-3-Kan
US3^deop^-3-Kan ide R	CGCGAGCCCATTTATACCCA	
PRV gI+gE’ F	GTCGTGGGCATCGTCATCAT	Amplification of gI and part of gE
PRV gI+gE’ R	TAGGAGATGGTACATCGCGG	
H1-US3^deop^-1-Kan-H2 F	CTTGCCGGGCTCAGCAGGGGGTTGTCGCGCGTCCACGCCCAGCGCTCGCACGCAGCAACA	Amplification of US3^deop^-1-Kan with homologous arms
H1-US3^deop^-1-Kan-H2 R	GTACAGATCGCACCGAAAGTGCGGCAGGACCAGGCACGTCACCGGGCCCCGGGCGAGCAT	
H1-US3^deop^-2-Kan-H2 F	CTGATGGAGGGCATGCTGCTGAAGCGCCTGGCCCACGATAACGTCATGAGCCTGAAGCAG	Amplification of US3^deop^-2-Kan with homologous arms
H1-US3^deop^-2-Kan-H2 R	GGGCGGGAACTCCTCGGGGTGCACCCCGCGGAGGGCGCGGATGAGGTCGATCAGGTGCAT	
H1-US3^deop^-3-Kan-H2 F	GAGACGCTGGCCTACCCCAAGACGATCACCGGCGGGGACGAGCCCGCGATCAACGGGGAG	Amplification of US3^deop^-3-Kan with homologous arms
H1-US3^deop^-3-Kan-H2 R	TTGTTGAGCTGTGGAGATGCGCAAAGGTGTGTGTGTCCTACCGCTCGGAGCCGGGCCGTT	
PRV US3 check F	AGCGCTCGCACGCAGCAACA	Identification of US3
PRV US3 check R	CTTTGGAATGTGGACCGTAT	
PRV ΔTK check F	CGCACCCCGAG GTTGACTTCAA	Identification of partial deletion of TK gene
PRV ΔTK check R	TTGTACGCGCCGAAGAGGGTGT	
PRV gE site check F	TCTGGAGGGGCCCT CGCCGA	Identification of gE
PRV gE site check R	AGAGAGAGGACGGAGGCGTGTCATC	
PRV US3^deop^-1/2 mRNA F	ATGCACCTGATCGACCTCAT	Real time PCR for determination of US3^deop^-1/2 mRNA
PRV US3^deop^-1/2 mRNA R	TTGTAAATCAGGAAAGCCCC	
PRV US3^deop^-3 mRNA F	GACACGGTGGTGCTGAAGGT	Real time PCR for determination of US3^deop^-1/2 mRNA
PRV US3^deop^-3 mRNA R	TACAGATCGCACCGAAAGTG	
US3^deop^-1 inred F	ATCTCGAGCTCAAGCTTCGAATTCATGGCCGATGCCGGAATCCCC	Amplification of US3^deop^-1, US3^deop^-2, and US3^deop^-3
US3^deop^-2/3 inred F	ATCTCGAGCTCAAGCTTCGAATTCATGGCCGACGCCGGAATCCC	
US3^deop^-1/2/3 inred R	GCGGTACCGTCGACTGCAGAATTCTACGGTCCACATTCCAAA	
pmKate2-US3 inred F	TTTAGTGAACCGTCAGATCC	Identification of US3^deop^-1/2/3 in pmKate2-N-US3^deop^-1, pmKate2-N-US3^deop^-2, and pmKate2-N-US3^deop^-3
pmKate2-US3 inred R	ACGGGATCAAACGTCAACAT	
gB F	GCGGTCACCTTGTGGTTGTT	Real time PCR for determination of gB mRNA
gB R	AACGTCATCGTCACGACCGT	

**Table 2 vaccines-11-01288-t002:** Data presenting the pathogenicity of different virus strains in mice.

Virus Strain	Inoculation Dose (TCID_50_/mL)	Quantity	Clinical Symptoms	Lethality
PRV^ΔTK&gE-US3deop−1^	10^7^	8	/	0
	10^6^	8	/	0
	10^5^	8	/	0
PRV^ΔTK&gE-US3deop−2^	10^7^	8	/	0
	10^6^	8	/	0
	10^5^	8	/	0
PRV^ΔTK&gE-US3deop−3^	10^7^	8	/	0
	10^6^	8	/	0
	10^5^	8	/	0
PRV^ΔTK&gEAH02^	10^7^	8	/	0
	10^6^	8	/	0
	10^5^	8	/	0
Control group	/	8	/	0

## Data Availability

The figures and tables supporting the results of this study are included in the article and Appendix A, and the original datasets are available from the first author or corresponding author upon request.

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
