# Peer review of "A Novel Strategy of US3 Codon De-Optimization for Construction of an Attenuated Pseudorabies Virus against High Virulent Chinese Pseudorabies Virus Variant"

_vaccines, 2023, doi:10.3390/vaccines11081288_

Round 1

Reviewer 1 Report

The present article reports about a novel strategy for construction of an attenuated pseudorabies virus by US3 codon de-optimization technique. The authors have tried to develop a new potential vaccine candidate against Pseudorabies infection through this strategy.

Following are the few comments after review of the paper;

1.     Line 26 – Why only 107.0 TCID50 dose was selected for immunization?

2.     Table 1 – Kindly modify the table with clearly mentioning the steps in which selected primers were used.

3.     Kindly mention the transfection technique used.

4.     Line no 180 – PCR and subsequently, sequencing is employed for verification. Kindly correct the same.

5.     How TCID50 and LD is calculated for the experiments? 

6.     Line 274-275 - Kindly elaborate the techniques used for the successful verification of deletion of genes and US3 codon de-optimization. 

7.     Line no 279 - Specify the injection volume of recombinants in mice?

8.     Kindly clarify why the vaccination is different in piglets for safety and immune protection test? Does it have any effect on data with respect to clinical manifestations, efficacy and toxicity?

9.     Which method has been used for detection of serological antibodies?  Kindly specify the neutralizing antibodies generated post immunization.

10.  Why the inoculation doses (TCID50) for mice and piglets are different to study safety and immunogenicity? Kindly clarify.

11.  Line no 460- Which group is PRVΔTK&PK&gE-AH02 in the study?

12.  Line No 463 – Please correct the spelling of “Second”.

13.  How detoxification and toxicity status detected? Kindly elaborate.

14.  Line no 531-535 – Kindly modify the lines as they are repetitive.

15.  General Comment 1 – Kindly cross - check the designation/name of recombinants as they are different in multiple places.

16.  General Comment 2 – Any reason behind not incorporating Barthak61 as one of the controls in study?

17.  General Comment 3 – Kindly present data in a proper tabulated format with each parameter clearly defined for study group in mice and piglets.

The article presented clearly demonstrated the strategy of codon de-optimization for preparation of attenuated pseudorabies virus. The strategy of TK & gE deletion have been reported earlier, however the present method, as described by the author, is novel. There are few limitations with respect to data presentation and the same can be modified following the reviewer's suggestions.

Author Response

Response to Reviewer 1 Comments

Point 1:  Line 26 – Why only 107.0 TCID50 dose was selected for immunization?

Response 1: Thank you for your question. In our previous study (Xu et al., 2022), we inoculated mice with different doses (107.0 TCID50, 106.0 TCID50 and 105.0 TCID50) to evaluate the safety of the PRV strain in mice and calculate the LD50 of the strain. The results showed that there were no clinical symptoms after vaccination with 107.0 TCID50. Furthermore, immunization with 107.0 TCID50 PRV TK/gE deletion strain or PRV TK/PK/gE deletion in mice can achieve ideal immune protection effects against PRV variant challenge. Therefore, 107.0 TCID50 dose was selected for immunization in the study.

We have now rephrased the sentence in the revision: Based on our previous study[18], after 2 weeks of immunization in the 107.0 TCID50 dose group, mice were administered 100 LD50 PRV AH02LA strain while an attack control group was established.

References cited here are as follows:

Xu, M., Zhang, C., Liu, Y., Chen, S., Zheng, Y., Wang, Z., Cao, R., Wang, J., 2022. A noval strategy of deletion in PK gene for construction of a vaccine candidate with exellent safety and complete protection efficiency against high virulent Chinese pseudorabies virus variant. Virus Res. 313, 198740.

Point 2: Table 1 – Kindly modify the table with clearly mentioning the steps in which selected primers were used.

Response 2: Thank you for the suggestion. We have modified Table 1 with clearly mentioning the steps/aims in which selected primers were used.

Point 3: Kindly mention the transfection technique used.

Response 3: Thank you for the suggestion. We used the Lipofectamine® 3000 transfection kit for transfection and added corresponding content to the manuscript. “Lipofectamine® 3000 was purchased from Invitrogen (Waltham, USA).”(Line 105) “ using Lipofectamine® 3000 following manual of supplier”(Line 209-210)

Point 4: Line no 180 – PCR and subsequently, sequencing is employed for verification. Kindly correct the same.

Response 4: Thanks for your reminding. We have changed “by PCR sequencing”to “by PCR and subsequently, sequencing.”(Line 168-169)

Point 5: How TCID50 and LD is calculated for the experiments? 

Response 5: TCID50 refers to the amount of virus that can cause cytopathic changes in half of the cell culture plates. After diluting the virus solution tenfold, inoculate the virus with different dilutions onto monolayer cells cultured on a 96 well plate. After a certain period of time, the cellular lesion was observed, and TCID50 was calculated according to the Reed-Muench formula. we have now rephrased the sentence in the revision as “The virus suspension was then harvested, and the virus titer was determined based on a 50% tissue culture infectious dose (TCID50) according to the Reed-Muench formula.” (Line 260-261).

LD50 refers to the minimum amount of virus required to cause half of the deaths of an animal through a certain infection pathway. After diluting the virus solution tenfold, inoculate the animals with different dilutions of the virus, and calculate the death rate of the animals within the specified time using the Reed-Muench formula to calculate LD50.

Point 6: Line 274-275 - Kindly elaborate the techniques used for the successful verification of deletion of genes and US3 codon de-optimization. 

Response 6: Thanks for your advice. Indeed, deletion of genes and US3 codon de-optimization were identified by PCR and subsequently, sequencing. we have now rephrased the sentence in the revision as “gE, TK and the recoded US3 genes in PRVΔTK&gE-US3deop-1, PRVΔTK&gE-US3deop-2, and PRVΔTK&gE-US3deop-3 were identified by PCR and sequencing with primers PRV ΔTK check F/R, PRV gE site check F/R, and PRV US3 check F/R, respectively. Finally, we observed whether the deletion of genes and US3 codon de-optimization was stably inherited.”(Line 265-269)

Point 7: Line no 279 - Specify the injection volume of recombinants in mice?

Response 7: Thanks for your reminding. We have supplemented the injection volume of recombinants in mice in the manuscript. “ with 0.2 mL of ” (Line 272-273)

Point 8: Kindly clarify why the vaccination is different in piglets for safety and immune protection test? Does it have any effect on data with respect to clinical manifestations, efficacy and toxicity?

Response 8: Thank you for the comment. In our previous study (Xu et al., 2022), piglets vaccinated with 105 TCID50 PRV TK/gE deletion strain or PRV TK/PK/gE deletion variant strain have no clinical symptoms, temperature response or virus shedding after AH02LA challenge. Therefore, 105 TCID50 were selected for immune protection test. As previously described (Wang et al., 2022; Xu et al., 2022), high-dose vaccination was using to evaluate the safety of the virus strain on piglets in clinical application. Therefore, vaccination is different in piglets for safety and immune protection test. We have now added some descriptions in the revision: High-dose vaccination was using to evaluate the safety of a live PRV vaccine on piglets in clinical application. (Line 298-300)

Our previous study showed that weaned piglets vaccinated with PRV TK/gE deletion strain with 105 TCID50 were protected against lethal challenge (Wang et al., 2022). No fever or other clinical symptoms were observed. Furthermore, piglets inoculated with 106 TCID50 PRV TK/gI/gE deletion strain remained healthy, without fever and clinical signs after challenge (Lv et al., 2021). Therefore, we speculate that different vaccination doses (105 TCID50 or 106 TCID50) may have no effect on data with respect to clinical manifestations, efficacy and toxicity in weaned piglets. However, newborn piglets are highly susceptible to PRV infection. In this study, 4 of 5 newborn piglets inoculated with 106 TCID50 PRV TK/gE deletion strain displayed a body temperature of 40-41℃ for 3~6 days, and 2 of them showed clinical symptoms, such as sneezing, coughing, and loss of appetite. In our previous study (Xu et al., 2022), 2 of 5 piglets inoculated with 106.5 TCID50 PRV TK/gE deletion strain showed typical clinical signs, one of the sick piglets died on the third day and the other died on the fifth day after inoculation. The body temperatures of all survived piglets reached over 40.0℃ for 5~7 days and one of the piglets had a body temperature of 41.1℃ on the fourth day after inoculation. Therefore, different vaccination doses can lead to inconsistent data on clinical manifestations and toxicity in newborn piglets.

References cited here are as follows:

Lv, L., Liu, X., Jiang, C., Wang, X., Cao, M., Bai, J., Jiang, P., 2021. Pathogenicity and immunogenicity of a gI/gE/TK/UL13-gene-deleted variant pseudorabies virus strain in swine. Vet. Microbiol. 258, 109104.

Wang, J., Song, Z., Ge, A., Guo, R., Qiao, Y., Xu, M., Wang, Z., Liu, Y., Zheng, Y., Fan, H., Hou, J., 2018. Safety and immunogenicity of an attenuated Chinese pseudorabies variant by dual deletion of TK&gE genes. BMC. Vet. Res. 14, 287.

Xu, M., Zhang, C., Liu, Y., Chen, S., Zheng, Y., Wang, Z., Cao, R., Wang, J., 2022. A noval strategy of deletion in PK gene for construction of a vaccine candidate with exellent safety and complete protection efficiency against high virulent Chinese pseudorabies virus variant. Virus Res. 313, 198740.

Point 9: Which method has been used for detection of serological antibodies?  Kindly specify the neutralizing antibodies generated post immunization.

Response 9: Thank you for the comment. We used fixed serum dilution virus neutralization test to detect neutralizing antibody in serum, and the result represents the neutralization index of serum. We have now added some descriptions for neutralization index of serum in the revision: 100 μL of each serum sample (heat inactivated for 30 min at 56 °C) was mixed with an equal volume of virus (AH02LA) at different dilution (TCID50). The neutralization indexes were expressed as the TCID50 of piglet serum in the test group divided by the TCID50 of negative serum. (Line 303-307)

We have specified the neutralizing antibody generated post immunization in the manuscript. “After 7 days of immunization, the serum antibody neutralizing index was low in PRVΔTK&gE-US3deop-1 and PRVΔTK&gE-AH02 treatment group, subsequently, increased progressively. After 14 days of immunization, the serum antibody neutralizing index was 50118 in PRVΔTK&gE-US3deop-1 treatment group and 72443 in PRVΔTK&gE-AH02 treatment group. After 21 days of immunization, the serum antibody neutralizing index was 1698244 in PRVΔTK&gE-US3deop-1 group and 2884031 in PRVΔTK&gE-AH02 group. Notably, no significant difference in the serum antibody indices was observed between the PRVΔTK&gE-AH02 and PRVΔTK&gE-US3deop-1 groups after 7, 14, and 21 days of immunization. In the later stage, we will also supplement the duration of antibody production in piglets after immunization with PRVΔTK&gE-US3deop-1.” (Line 457-466)

Point 10: Why the inoculation doses (TCID50) for mice and piglets are different to study safety and immunogenicity? Kindly clarify.

Response 10: Thanks for your question. In our laboratory, we have established different challenge models of PRV AH02LA strain in mice and pigs. In the PRV challenge model of mice, the vaccine can achieve effective immune protection against ultra-high challenge doses. Generally, a challenge dose of 100LD50 is selected to evaluate the immune protection effect of vaccine against PRV. In the PRV challenge model of pigs, the challenge dose of approximately 2LD50 is mostly selected, which can not only ensure the establishment of the challenge control group, but also effectively evaluate the immune efficacy of the vaccine against PRV.

Point 11: Line no 460- Which group is PRVΔTK&PK&gE-AH02 in the study?

Response 11: We apologize for our carelessness and thanks for your reminding. We have changed “PRVΔTK&PK&gE-AH02” to “PRVΔTK&gE-US3deop-1”. (Line 464)

Point 12:  Line No 463 – Please correct the spelling of “Second”.

Response 12: We apologize for our carelessness and thanks for your reminding. We have changed “second” to “second”. (Line 468)

Point 13: How detoxification and toxicity status detected? Kindly elaborate.

Response 13: We apologize for our inappropriate description. We have changed “detoxification” to “prevent virus shedding”. Nasal swab samples were collected and weighed daily from 0 to 14 dpc to virus shedding. After shaking and freeze-thaw cycles (-80°C and 37°C), samples were centrifuged (10, 000 rpm) and the supernatants were used to determine the viral titers. We have now added some descriptions for virus shedding in the revision: Nasal swab samples were collected and weighed daily from 0 to 14 dpc to detect detoxification. After shaking and freeze-thaw cycles (-80°C and 37°C), samples were centrifuged (10, 000 rpm) and the supernatants were used to determine the viral titers. (Line 316-319). Furthermore, we have now rephrased the sentence in the revision: Nasal swabs were collected daily in the challenge-infected to determine virus titers in the excreted nasal discharge. It was observed that all piglets in the control group shed virus starting on 1 day after the attack and continued to do so until death. To this end, no virus was detected after the challenge in PRVΔTK&gE-US3deop-1 and PRVΔTK&gE-AH02 immunized treatment groups (Figure 6C and Supplementary Tables S4). (Line 483-487)

Point 14: Line no 531-535 – Kindly modify the lines as they are repetitive.

Response 14: Thanks for your reminding. we have now rephrased the sentence in the revision: Further, a piglet safety test showed that PRVΔTK&gE-US3deop-1 was less pathogenic in piglets than PRVΔTK&gE-AH02, producing high levels of neutralizing antibodies. Finally, evaluation of the immunoprotective effect of PRVΔTK&gE-US3deop-1 in piglets showed that it not only provided complete protection against the highly virulent variant AH02LA challenge after 1 week of immunization but also prevent virus shedding. (Line 538-543)

Point 15: General Comment 1 – Kindly cross - check the designation/name of recombinants as they are different in multiple places.

Response 15: We appreciated your attention to detail. We carefully checked the manuscript and corrected the designation/name of recombinants.

Point 16: General Comment 2 – Any reason behind not incorporating Barthak61 as one of the controls in study?

Response 16: Thanks for your question. Our previous studies showed that pseudorabies virus TK/gE double gene deletion strain(PRVΔTK&gE-AH02)can provide better immune protection against AH02LA compared to Bartha-k61 in piglets (Wang et al., 2018). Therefore, in this study, PRVΔTK&gE-AH02 was used as the control group to evaluate the immune protective effect of PRVΔTK&gE-US3deop-1 in piglets.

References cited here are as follows:

Wang, J., Song, Z., Ge, A., Guo, R., Qiao, Y., Xu, M., Wang, Z., Liu, Y., Zheng, Y., Fan, H., Hou, J., 2018. Safety and immunogenicity of an attenuated Chinese pseudorabies variant by dual deletion of TK&gE genes. BMC Vet Res 14(1), 287.

Point 17: General Comment 3 – Kindly present data in a proper tabulated format with each parameter clearly defined for study group in mice and piglets.

Response 17: We have added a table based on your suggestion, including each parameter clearly defined for study group in mice and piglets, which is included in the supplementary file.

Reviewer 2 Report

The article represents an interesting approach to the construction of a pseudo-rabies vaccine. The article shows several constructs the authors claim efficiently induce immune response without any toxic effects. There are, however, some points which are unclear and need to be addressed. The efficiency of the electroporation of the different constructs has not been detailed, and it is important due to the sequences shown. The authors claim that PCR identified the constructs; why not use sequencing to show the validity size and proper orientation? Important details are also missing in Figure 2C, Figures 3 b and 3 C, and 4-6 (statistics n=?). The authors did not show why, despite the stability of the constructs, the amount of neutralizing antibodies did not significantly increase after day 14 and how long they could detect them.   Figure 6 is difficult to understand after viewing Figure 5. The amount of replicating viruses is very high. The data should be included in the supplemental files. The discussion and conclusions should be modified. There are several sentences that require atention.

Several sentences should be modified. Minor typo errors were also found

Author Response

Response to Reviewer 2 Comments

Point 1: The efficiency of the electroporation of the different constructs has not been detailed, and it is important due to the sequences shown.

Response 1: Thank you for your advice. We have added relevant content on electroporation efficiency of different building blocks according to your suggestion. “Approximately 100-150 monoclonal colonies were grown on each plate. There is no significant difference among 3 constructs.” (Line 137-138)

Point 2: The authors claim that PCR identified the constructs; why not use sequencing to show the validity size and proper orientation?Important details are also missing in Figure 2C, Figures 3 b and 3 C, and 4-6 (statistics n=?).

Response 2: Thanks for your valuable advice. Indeed, deletion of genes and US3 codon de-optimization were identified by PCR and subsequently, sequencing. We have now rephrased the sentence in the revision as “gE, TK and the recoded US3 genes in PRVΔTK&gE-US3deop-1, PRVΔTK&gE-US3deop-2, and PRVΔTK&gE-US3deop-3 were identified by PCR and sequencing with primers PRV ΔTK check F/R, PRV gE site check F/R, and PRV US3 check F/R, respectively. Finally, we observed whether the deletion of genes and US3 codon de-optimization was stably inherited.”(Line 265-269)

We have added 'n=' in Figure 2C, Figures 3 b and 3 C, and 4-6 .

Point 3: The authors did not show why, despite the stability of the constructs, the amount of neutralizing antibodies did not significantly increase after day 14 and how long they could detect them.

Response 3: We apologize for our inappropriate description. we have now rephrased the sentence in the revision: After 7 days of immunization, the serum antibody neutralizing index was low in PRVΔTK&gE-US3deop-1 and PRVΔTK&gE-AH02 treatment group, subsequently, increased progressively. After 14 days of immunization, the serum antibody neutralizing index was 50118 in PRVΔTK&gE-US3deop-1 treatment group and 72443 in PRVΔTK&gE-AH02 treatment group. After 21 days of immunization, the serum antibody neutralizing index was 1698244 in PRVΔTK&gE-US3deop-1 group and 2884031 in PRVΔTK&gE-AH02 group. (Line 457-462)

Point 4: Figure 6 is difficult to understand after viewing Figure 5. The amount of replicating viruses is very high.The data should be included in the supplemental files.

Response 4: Thank you for the comment. Figure 5 shows the evaluation of the pathogenicity of PRVΔTK&gE-US3deop-1, while Figure 6 shows the piglet challenge protection test, which involves immunizing piglets with PRVΔTK&gE-US3deop-1 and then using the wild strain AH02LA for challenge to evaluate the immunogenicity of PRVΔTK&gE-US3deop-1. We have now revised Figure 6 in the revision.

Point 5: The discussion and conclusions should be modified. There are several sentences that require atention.

Response 5: Thanks for your suggestion. We have made modifications to the discussion and conclusion sections.

Round 2

Reviewer 2 Report

The manuscript has been partially improved; however, some issues remain to be discussed, as preventing viral shedding by vaccinating with one construct is not convincing. Figure 3 parts and C can not be detailed properly. There are some grammatical mistakes in the new text that has to be corrected 

There are still grammatical errors in the manuscript that require correction.

Author Response

Response to Reviewer 2 Comments

Point 1: The manuscript has been partially improved; however, some issues remain to be discussed, as preventing viral shedding by vaccinating with one construct is not convincing. Response 1: Thank you for your advice. In the study, evaluation of the immunoprotective effect of PRVΔTK&gE-US3deop-1 in piglets showed that it not only provided complete protection against the highly virulent variant AH02LA challenge but also prevented virus shedding at 1 week post immunization. It is clear that levels of antibodies correlate poorly with the decreased virus replication early after infection, PRV specific lymphocyte proliferation responses and a rapid influx of T lymphocytes at the site of viral replication plays an important role in the clearance of a PRV infection(Bouma et al., 1997; van Rooij et al., 2004). Therefore, PRV specific cell-mediated immune response may been related to the prevention of clinical disease and virus shedding early after challenge. Future studies involving cell-mediated immunity analysis are necessary to better understand the mechanism of immune protection induced by live vaccines.

We have now added some descriptions in the revision: It is clear that levels of antibodies correlate poorly with the decreased virus replication early after infection, PRV specific lymphocyte proliferation responses and a rapid influx of T lymphocytes at the site of viral replication plays an important role in the clearance of a PRV infection(Bouma et al., 1997; van Rooij et al., 2004). Therefore, PRV specific cell-mediated immune response may been related to the prevention of clinical disease and virus shedding early after challenge. Future studies involving cell-mediated immunity analysis are necessary to better understand the mechanism of immune protection induced by live vaccines. (Line 498-504)

References cited here are as follows:

Bouma, A., Zwart, R.J., De Bruin, M.G., De Jong, M.C., Kimman, T.G., Bianchi, A.T., 1997. Immunohistological characterization of the local cellular response directed against pseudorabies virus in pigs. Vet Microbiol 58, 145-154.

van Rooij, E.M., de Bruin, M.G., de Visser, Y.E., Middel, W.G., Boersma, W.J., Bianchi, A.T., 2004. Vaccine-induced T cell-mediated immunity plays a critical role in early protection against pseudorabies virus (suid herpes virus type 1) infection in pigs. Vet Immunol Immunopathol 99, 113-125.

Point 2: Figure 3 parts and C can not be detailed properly. 

Response 2: Thanks for your valuable advice. We have now added some descriptions in the Figure 3.

Point 3: There are some grammatical mistakes in the new text that has to be corrected

Response 2: Thanks for your valuable advice. We have carefully checked the entire manuscript, and some grammatical mistakes have been revised.

Round 3

Reviewer 2 Report

The manuscript has been modified accordingly. 

Minor grammar mistakes are observed.